# Impact of Antibiotic Authorisation at Three Provincial Hospitals in Thailand: Results from a Quasi-Experimental Study

**DOI:** 10.3390/antibiotics11030354

**Published:** 2022-03-07

**Authors:** Walaiporn Wangchinda, Jintana Srisompong, Sunee Chayangsu, Darat Ruangkriengsin, Visanu Thamlikitkul, Pornpan Koomanachai, Rujipas Sirijatuphat, Pinyo Rattanaumpawan

**Affiliations:** 1Department of Medicine, Division of Infectious Diseases and Tropical Medicine, Faculty of Medicine Siriraj Hospital, Mahidol University, Bangkok 10700, Thailand; walaiporn.wan@mahidol.ac.th (W.W.); visanu.tha@mahidol.ac.th (V.T.); pornpan.koo@mahidol.ac.th (P.K.); rujipas.sir@mahidol.ac.th (R.S.); 2Internal Medicine Unit, Surat Thani Hospital, Surat Thani 84000, Thailand; jint9839@gmail.com; 3Internal Medicine Unit, Surin Hospital, Surin 32000, Thailand; chayangsu.sunee@gmail.com; 4Internal Medicine Unit, Sa Kaeo Crown Prince Hospital, Sa Kaeo 27000, Thailand; darat_r@yahoo.com

**Keywords:** antimicrobial stewardships, antibiotic authorization, drug use evaluation

## Abstract

Implementing antimicrobial stewardship (AMS) at non-university hospitals is challenging. A quasi-experimental study was conducted to determine the impact of customised antibiotic authorisation implementation on antimicrobial consumption and clinical outcomes at three provincial hospitals in Thailand. Customised pre-authorisation of selected restricted antibiotics and post-authorisation of selected controlled antibiotics were undertaken and implemented at each hospital by the local AMS team with guidance from the AMS team at the university hospital. From January 2019–December 2020, there were 1802 selected patients (901 patients during the pre-implementation period and 901 patients during the post-implementation period). The most commonly used targeted antimicrobial was meropenem (49.61%), followed by piperacillin/tazobactam (36.46%). Comparison of the outcomes of the patients during the pre- and post-implementation periods revealed that the mean day of therapy of the targeted antimicrobials was significantly shorter during the post-implementation period (6.24 vs. 7.64 days; *p* < 0.001), the favourable clinical response (the improvement in all clinical and laboratory parameters at the end of antibiotic therapy) was significantly higher during the post-implementation period (72.70% vs. 68.04%; *p* = 0.03) and the mean length of hospital stay was significantly shorter during the post-implementation period (15.78 vs. 18.90 days; *p* < 0.001). In conclusion, implementation of antibiotic authorisation at provincial hospitals under experienced AMS team’s guidance was feasible and useful. The study results could be a good model for the implementation of customised AMS strategies at other hospitals with limited resources.

## 1. Introduction

In 2016, Thailand’s national strategic plan on antimicrobial resistance (AMR) was developed in accordance with the global action plan on AMR launched by the World Health Organization in 2015 [1]. One of the six strategic actions in Thailand’s national strategic plan on AMR is to improve antimicrobial use and infection prevention and control (IPC) at hospitals. The antimicrobial stewardship program (ASP) has been demonstrated to promote appropriate antimicrobial use, reduce antimicrobial consumption, improve clinical outcomes, and reduce the emergence of AMR without compromising the patients’ outcomes in terms of infection-related morbidity and mortality [2,3,4,5,6,7,8]. Pre-prescription authorisation and post-prescription authorisation of selected antimicrobials are considered as the keys of antimicrobial stewardship (AMS) strategies and are strongly recommended by the Infectious Diseases Society of America and the Society for Healthcare Epidemiology of America [9].

A recent nationwide survey of 399 acute-care hospitals in Thailand revealed that most hospitals in Thailand had an ASP in place. However, less than one-quarter of these hospitals had implemented an antibiotic authorisation strategy. The major obstacles of ASP implementation identified in this survey were increased workload, lack of AMS knowledge and skills of relevant personnel, and lack of hospital administrator concern [10]. Furthermore, an affiliation with a university hospital was found to be an independent factor associated with successful ASP implementation [11]. Therefore, customised AMS strategies that are specifically designed for individual hospitals on the basis of hospital infrastructure, epidemiology, local information, and capability of personnel should be promoted.

Given these considerations, a quasi-experimental (pre- and post-intervention) study to evaluate the impact of customised AMS strategies implementation under the support of the AMS team at the university hospital was conducted.

## 2. Results

### 2.1. Baseline Characteristics of Patients in the Pre-Implementation and Post-Implementation Periods

A total of 1802 patients from three participating hospitals were included, among whom 901 patients were in the pre-implementation period, and 901 patients were in the post-implementation period. Characteristics of the patients in both periods are shown in Table 1.

Most patients were male (56.27%) with mean age of 60.61 ± 16.31 years. The mean ± SD length of hospital stay was 17.35 ± 17.23 days. Most patients were admitted to internal medicine wards (73.47%) and had at least one comorbidities (85.57%). Intensive care unit (ICU) admission was observed in 30.85% of patients. Almost half of the patients received at least one medical intervention, such as central venous catheter, urinary catheter, or nasogastric tube. Sixty-two percent of patients had previously been exposed to at least one antimicrobial within 3 months, and the cephalosporin group was the most common agent (54.16%) that such patients received. Previous colonisation with multidrug resistant (MDR) bacteria was identified in 10.77%, and the most common colonised bacteria was extended spectrum beta-lactamase (ESBL) producing-*Enterobacterales* or ceftriaxone-resistant *Enterobacterales* (7.10%). The patients in the post-implementation period were significantly older (61.92 ± 16.38 years vs. 59.30 ± 16.15 years; *p* < 0.001), less likely to undergo the placement of a central line catheter prior to the hospitalization (4.99% vs. 7.55%; *p* = 0.03), more likely to undergo the placement of a urinary catheter prior to the hospitalization (51.94% vs. 31.74%; *p* < 0.001), more likely to have previous antimicrobial use (70.70% vs. 53.39%; *p* < 0.001), and more likely to have MDR bacterial colonisation (12.54% vs. 8.99%; *p* = 0.02) than those in the pre-implementation period. However, patients in the post-implementation period had a lower rate of ICU admission (26.75% vs. 34.96%; *p* < 0.001) than those in the pre-implementation period.

### 2.2. Details of Infection and Antimicrobial Therapy of Patients in the Pre-Implementation and Post-Implementation Periods

Details of infection and antimicrobial therapy of patients in the pre- and post-implementation periods are shown in Table 2. The most commonly used targeted antimicrobial was meropenem (49.61%), followed by piperacillin/tazobactam (36.46%). The main indication of these prescriptions was for the treatment of hospital-acquired infections (71.86%). The three leading sites of infection were lower respiratory tract (40.23%), urinary tract (23.86%), and bloodstream (21.03%). The most common causative pathogen was *Escherichia coli* (18.04%).

Patients in the post-implementation period had a higher proportion of antibiotic prescriptions for the treatment of hospital-acquired infections (78.25% vs. 65.48%; *p* < 0.001) than those in the pre-implementation period. Distribution of the sites of infection and causative pathogens of the patients in both periods were similar, except patients in the pre-implementation period had a significantly higher proportion of lower respiratory tract infections (44.51% vs. 35.96%; *p* < 0.001) and *Acinetobacter baumannii* infections (13.76% vs. 9.88%; *p* = 0.01) than those in the post-implementation period.

The mean day of therapy (DOT) of targeted antimicrobials in patients in the post-implementation period was also significantly shorter than that in patients in the pre-implementation period (6.24 ± 4.84 days vs. 7.64 ± 6.10 days; *p* < 0.001). DOT of all antimicrobials during the pre-implementation and post-implementation periods are shown in Appendix A. The rates of ceftazidime prescription (10.43% vs. 4.55%.; *p* < 0.001) and other cephalosporin prescriptions (15.43% vs. 7.55%; *p* < 0.001) were significantly higher in patients in the post-implementation period than those in the pre-implementation period.

The clinical outcomes of patients in the pre- and post- implementation periods are shown in Table 3. Patients in the post-implementation period had a significantly higher favourable clinical response (72.70% vs. 68.04%; *p* = 0.03) and a shorter mean length of hospital stay (15.78 days vs. 18.90 days; *p* < 0.001) than those in the pre-implementation period. Bacterial superinfection was less likely in patients in the post-implementation period than in those in the pre-implementation period (7.21% and 12.10%; *p* < 0.001). However, there was no significant difference in antibiotic-associated diarrhoea or in-hospital mortality between the patients in both periods.

### 2.3. Factors Associated with Favourable Clinical Response at the End of Antibiotic Therapy

Results of multivariate analysis to identify factors associated with favourable clinical response at the end of antibiotic therapy are shown in Table 4. Patients in the post-implementation period, hospitalisation for elective surgery, and having infection caused by *E. coli* were associated with favourable clinical response at the end of antimicrobial therapy whereas patients with underlying cardiovascular disease, having an immunocompromised status, having acute kidney injury, and having infection caused by *A. baumannii* were associated with unfavourable clinical response at the end of antimicrobial therapy.

### 2.4. Evaluation of Antimicrobial Prescriptions for Patients in the Post-Implementation Period

Reasons for antimicrobial prescriptions during the post-implementation period that were retrieved from antimicrobial request and approval (ATB) forms are shown in Table 5. Among 901 prescriptions of the targeted antimicrobials in the post-implementation period, 5.55% were pre-authorisation prescriptions, and 94.45% were post-authorisation prescriptions. Pre-authorisation prescriptions were all prior approved by the AMS team. Post-authorisation evaluation was performed in 739 prescriptions. Of these, 89.99% were approved and 10.01% were not approved. Approximately one-quarter (13.16%) of prescriptions were automatically discontinued in patients who expired or were voluntarily modified by the attending physician before evaluation by the AMS team was made. The three leading additional recommendations provided by the AMS team were source control of infection (3.77%), additional laboratory or imaging investigation (3.33%), and switching from parenteral administration of antibiotics to oral administration (2.33%).

## 3. Discussion

Pre-prescription authorisation and post-prescription authorisation of selected antimicrobials are important AMS strategies that have been widely recommended as the foundational interventions for ASP at hospitals to promote appropriate antimicrobial use [2,9]. However, such interventions require the effort and dedication of relevant personnel. Furthermore, 24/7 availability of an antimicrobial authorisation consultation team is necessary for the implementation of a pre-prescription authorisation strategy. Therefore, implementation of antimicrobial authorisation at hospitals with limited human resources is extremely challenging. Most hospitals in Thailand usually have a high volume of patients, inadequate resources, and microbiology laboratories with limited capability. Most importantly, they usually have no personnel who are experienced in infectious diseases, such as infectious disease physicians or infectious disease pharmacists. Given these considerations, implementation of AMS strategies should take the aforementioned issues into consideration and should be customised according to the context of each hospital.

After implementation, the antimicrobial prescription pattern was remarkably changed. The targeted antimicrobial consumption in terms of DOT was significantly reduced. DOT was used in this study for overall measurement of the impact of ASP instead of defined daily dose (DDD) according to the recommendations of the Guidelines by the Infectious Diseases Society of America and the Society for Healthcare Epidemiology of America [6]. Furthermore, targeted antimicrobials were more likely to be prescribed for the treatment of hospital-acquired infections during the post-implementation period than the pre-implementation period. Given that the targeted antimicrobials were broad-spectrum and should be reserved for the treatment of hospital-acquired infections, a higher proportion of prescriptions for the treatment of hospital-acquired infections during the post-implementation period may be a proxy of more appropriate prescriptions during such a period. Furthermore, cephalosporins were more frequently prescribed during the post-implementation period, which may be a result of the squeezing balloon effect [12].

On the basis of the study results, the patients in post-implementation period had a higher rate of favourable clinical response, shorter ICU stays, shorter ventilator dependency duration, shorter fever duration, shorter hospital stays, and a lower rate of bacterial superinfections. Although many observed outcomes of the patients during the post-implementation period were remarkably better than those during the pre-implementation period, the magnitude of difference in these outcomes seemed to be small. These observations might be owing to the fact that patients had more severe or complicated infections during the post-implementation period. Targeted antimicrobials were needed for these infections according to the findings that more patients during the post-implementation period had previous use of cephalosporins, previous colonisation and/or infection with ESBL-producing *Enterobacterales*, and more hospital-acquired infections than patients during the pre-implementation period. The magnitude of the observed outcomes might be larger if the aforementioned attributes were comparable between the pre- and post-implementation periods. However, the study results clearly demonstrated the feasibility and clinical benefits of customised antibiotic authorisation as a component of ASP in hospitals. Our study reported the success of customised antimicrobial authorisation strategies under the collaboration of experienced AMS teams and local AMS teams at each participating hospital. One of the reasons for this success is that all participating hospitals had at least one infectious disease physician and had full support from their hospital administrator.

The non-approval rate of targeted antimicrobial prescriptions in the present study was rather low (10.01%) when compared with the results from other studies (30–64%) [13,14]. The relatively lower non-approval rate may be explained by the unavailability of microbiological results for patients who received targeted antimicrobials. It would be difficult for AMS teams to adjust antimicrobial regimens without knowing the causative pathogen and the susceptibility results. Moreover, the study non-approval did not include all non-approved prescriptions of restricted antimicrobials that required pre-authorisation.

This study had some limitations. First, the in-depth details of antibiotic authorisation (i.e., reason for prescription, additional suggestions by the AMS team) were only collected during the post-implementation period. During the pre-implementation period, Sakaeo Crown Prince Hospital and Surin Hospital did not have an official system of antibiotic authorisation, while Surat Thani Hospital did not have antibiotic authorisation. Second, the implementation period and post-implementation period occurred during the COVID-19 outbreak in Thailand. The COVID-19 situation may have had effects on the rate and type of hospitalisation, the hospital epidemiology of nosocomial infection, and the pattern of antimicrobial use at the hospitals. To avoid these confounders, we extended the wash-out period to the end of June 2020 when the incidence of COVID-19 cases was extremely low. Lastly, co-intervention and bias in outcome evaluation may be an issue because of the nature of the unblinded study.

Our study has several strengths. First, AMS strategies were specifically customised for each participating hospital according to hospital context. Second, we enrolled patients from three hospitals located in different geographic regions of Thailand. To reduce the potential bias that may occur because of the study design, we performed a multivariate analysis to determine whether the implemented AMS strategies were an independent factor associated with favourable outcomes.

In conclusion, two important AMS strategies (antibiotic authorisation and ATB forms) have been successfully implemented at three non-university hospitals in Thailand. The implementation of these AMS strategies not only reduced antimicrobial consumption, but also improved some clinical outcomes. The results from this study could be used as a good model for the implementation of customised AMS strategies at other hospitals with limited resources.

## 4. Material and Methods

### 4.1. Study Design and Setting

From January 2019 to December 2020, a quasi-experimental (pre- and post-intervention) study was conducted at three provincial hospitals in Thailand: Sakaeo Crown Prince Hospital, a 400-bed provincial hospital in Sakaeo Province, Eastern Thailand); Surin Hospital, a 900-bed provincial hospital in Surin Province, North-eastern Thailand, and Surat Thani Hospital, an 800-bed provincial hospital in Surat Thani Province, Southern Thailand. Details of hospital infrastructure, existing antimicrobial stewardship activities, and targeted antimicrobials at these participating hospitals are shown in Table 6.

This study was a component of the Expanded Antimicrobial Stewardship Project (Thailand Expanded ASP), which aimed to implement three important AMS activities: enhancement of antimicrobial resistance surveillance in isolated bacteria at microbiology laboratories by the global antimicrobial surveillance system (GLASS); development and implementation of clinical practice guidelines (CPGs) for antibiotic therapy of common infections; and building on the authorisation of selected antimicrobial agents at these participating hospitals. Details of the Thailand Expanded ASP are available elsewhere [9].

The study protocol was approved by the Institutional Review Board (IRB) of the Faculty of Medicine Siriraj Hospital, Mahidol University, Bangkok, Thailand (Certificate of approval number 384/2019) and the IRBs of all three participating hospitals. Waiver of informed consent from patients was granted because such AMS activities were considered part of the quality-of-care improvement program.

### 4.2. Study Subjects

Eligible patients were hospitalised patients aged ≥18 years who had received at least one dose of the targeted antimicrobials. The targeted antimicrobials were restricted and controlled antibiotics that were different among the participating hospitals. They usually included beta-lactam plus beta-lactamase inhibitor (ceftolozane/tazobactam and piperacillin/tazobactam), carbapenems (biapenem, ertapenem, imipenem/cilastatin, meropenem), colistin, sulbactam, tigecycline, and vancomycin.

The anticipated number of targeted antimicrobial prescriptions was 1000 prescriptions per year per hospital. Therefore, we randomly sampled at least 300 targeted antimicrobial prescriptions for each hospital. A total of 600 hundred patients (300 patients during the pre-implementation period and 300 patients during the post-implementation period) were randomly selected from eligible patients at each participating hospital. If the selected patients had more than one hospitalisation, only the first hospitalisation of such patients was included.

### 4.3. Study Procedures

The study was divided into three periods:(1)Pre-implementation period (January–December 2019)

Siriraj Hospital AMS team visited each participating hospital to gather baseline hospital information including hospital infrastructure, local antibiograms of the isolated bacteria in 2019, existing AMS activities, and list of available antimicrobials. No additional intervention was implemented during this period;
(2)Implementation and washout period (January–June 2020)

After obtaining the aforementioned baseline information, a strategic planning meeting involving both the Siriraj Hospital AMS team and the local AMS multidisciplinary team was held at each participating hospital. The antibiotic authorisation strategy at each participating hospital was individually customised on the basis of hospital epidemiology (distribution and susceptibility pattern of causative pathogens) and the list of available antimicrobials at each participating hospital.

The hospital antimicrobial list was divided into three categories: general antimicrobials, restricted antimicrobials, and controlled antimicrobials. General antimicrobials were those that could be prescribed by any physician without approval. Restricted antimicrobials were those that required approval by an AMS physician before prescribing the first dose. Controlled antimicrobials were those that could be prescribed by any physician for a short period of time (i.e., within 5 days) and continuation of such antimicrobials could be made only after approval by an AMS physician. If the targeted antimicrobial was inappropriately prescribed, it had to be discontinued. An alternative antimicrobial regimen could be suggested if necessary. The antimicrobial request and approval form for restricted and controlled antimicrobials (ATB forms) was also developed for use at each hospital. The AMS team could provide additional suggestions in the ATB form as necessary (i.e., dose adjustment, monitoring adverse reaction, necessary investigation, or intervention).

This was the first time that Sakaeo Crown Prince Hospital and Surin Hospital implemented an antibiotic authorisation strategy and the use of these ATB forms, whereas Surat Thani Hospital had used the ATB form for some antimicrobials but had never implemented an antibiotic authorisation strategy.

Two sequential meetings were held at each participating hospital. The first meeting was an operational meeting involving only the multidisciplinary AMS team and relevant personnel for antibiotic authorisation. The second meeting included educational sessions and workshops focusing on rational antimicrobial use and the details of new or additional antibiotic authorisation strategies. All healthcare personnel of the participating hospitals were invited to attend the second meeting.

The wash-out period was from January to June 2020 after the cessation of the first wave of the COVID-19 outbreak in Thailand.
(3)Post-implementation period (July–December 2020)

The antibiotic authorisation strategy and the use of ATB forms was fully implemented at all three participating hospitals. Siriraj Hospital AMS team was available for consultation by the local AMS teams at participating hospitals to resolve any onsite problems that may occur.

### 4.4. Data Collection

All necessary data were collected from the medical records of the selected patients who were prescribed targeted antimicrobials. Antimicrobial consumption in terms of type and day of therapy (DOT) was retrieved from the hospital pharmacy database. ATB forms were subsequently evaluated to determine indications of antimicrobial prescription, appropriateness of antimicrobial use, and additional recommendations by the hospital AMS teams. The ATB forms were only available during the post-implementation period.

### 4.5. Outcomes of Interest

The primary outcome was antimicrobial consumption in terms of DOT of targeted antimicrobials and all antimicrobials. Secondary outcomes included clinical response of the patient at the end of antimicrobial therapy, microbiological response of culture-proven infection, in-hospital mortality, and length of hospital stay (LOS).

Favourable clinical response was defined as complete recovery or improvement of signs and symptoms and laboratory results related to the index infection. Favourable microbiological response was defined as an absence of the original baseline pathogen during or following a course of antimicrobial therapy. If a given patient did not have a follow-up culture, they were not counted as a denominator.

Bacterial superinfection was defined as a new episode of bacterial infection that occurred within 4 weeks after the onset of index of infection.

### 4.6. Statistical Analysis

Categorical variables were reported as frequencies and percentages, while continuous variables were reported as means (standard deviations) or medians (ranges) as appropriate. Data of the patients during the pre- and post-implementation periods were compared. Chi-squared or Fisher exact test was used to compare categorical variables, while independent *t*-test or Mann–Whitney *U* test was used to compare continuous variables. Multivariate analysis was subsequently performed to identify associated factors for favourable clinical outcomes. All statistical analyses were performed using Stata version, 14.0 (Stata Corp, College Station, TX, USA) with two-sided analysis. A *p*-value of 0.05 or less was considered statistically significant.

## Figures and Tables

**Table 1 antibiotics-11-00354-t001:** Characteristics of patients in the pre- and post-implementation periods.

Variables	Total *n* = 1802 no (%)	Pre *n* = 901 no (%)	Post *n* = 901 no (%)	*p*-Value
**Age, mean ± SD, years**	60.61 ± 16.31	59.30 ± 16.15	61.92 ± 16.38	<0.001
**Male gender**	1014 (56.27)	516 (57.27)	498 (55.27)	0.39
**Hospital site**				0.94
Sakaeo Crown Prince hospital	594 (32.96)	294 (32.63)	300 (33.30)	
Surin hospital	600 (33.30)	300 (33.30)	300 (33.30)	
Surat Thani hospital	608 (33.74)	307 (34.07)	301 (33.41)	
**Ward type**				
General ward	1246 (69.15%)	586 (65.04%)	660 (73.25)	<0.001
Intensive care unit	556 (30.85)	315 (34.96)	241 (26.75)	
**Department**				
Medicine	1324 (73.47)	662 (73.47)	662 (73.47)	0.70
Surgery	395 (21.92)	194 (21.53)	201 (22.31)	
Other	83 (4.61)	45 (4.99)	38 (4.22)	
**Having at least one comorbid condition**	1542 (85.57)	768 (85.24)	774 (85.90)	0.69
Hypertension	738 (40.95)	363 (40.29)	375 (41.62)	0.57
Cerebrovascular diseases	320 (17.76)	154 (17.09)	166 (18.42)	0.46
Respiratory tract diseases *	203 (11.27)	100 (11.10)	103 (11.43)	0.82
Cerebrovascular diseases ^¥^	264 (14.65)	99 (10.99)	165 (18.31)	<0.001
Diabetes mellitus	499 (27.69)	242 (26.86)	257 (28.52)	0.43
Renal diseases ^Φ^	287 (15.93)	148 (16.43)	139 (15.43)	0.56
Hepatic diseases	225 (12.49)	119 (13.21)	106 (11.76)	0.35
Haematological diseases	70 (3.88)	41 (4.55)	29 (3.22)	0.14
Malignancy	301 (16.70)	131 (14.54)	170 (18.87)	0.01
Post-transplantation	4 (0.22)	3 (0.33)	1 (0.11)	0.32
Immunocompromised host ^Ψ^	163 (9.05)	78 (8.66)	85 (9.43)	0.57
HIV disease	60 (3.33)	39 (4.33)	21 (2.33)	0.02
Underwent placement of catheter prior to hospitalization				
– Central line catheter	113 (6.27)	68 (7.55)	45 (4.99)	0.03
– Urinary catheter	754 (41.84)	286 (31.74)	468 (51.94)	<0.001
**Previous exposure to antimicrobial agent within the past 3 months** ^Ω^	1118 (62.04)	481 (53.39)	637 (70.70)	<0.001
Penicillins	73 (4.05)	43 (4.77)	30 (3.33)	0.12
Cephalosporins	976 (54.16)	401 (44.51)	575 (63.82)	<0.001
Carbapenems	150 (8.32)	62 (6.88)	88 (9.77)	0.03
Beta-lactam/beta-lactamase inhibitors	207 (11.49)	102 (11.32)	105 (11.65)	0.83
Fluoroquinolones	107 (5.94)	45 (4.99)	62 (6.88)	0.09
Others	390 (21.64)	194 (21.53)	196 (21.75)	0.91
**Previous colonisation and/or infection with multidrug-resistant (MDR) organism**	194 (10.77)	81 (8.99)	113 (12.54)	0.02
MDR *A. baumannii*	42 (2.33)	21 (2.33)	21 (2.33)	1.00
MDR *P. aeruginosa*	21 (1.17)	11 (1.22)	10 (1.11)	0.83
Extended spectrum beta-lactamase-producing *Enterobacterales*	128 (7.10)	52 (5.77)	76 (8.44)	0.03
Carbapenem-resistant *Enterobacterales*	18 (1.00)	11 (1.22)	7 (0.78)	0.34
Methicillin-resistant *S. aureus*	4 (0.22)	3 (0.33)	1 (0.11)	0.32
Others	7 (0.39)	5 (0.55)	2 (0.22)	0.26

* Chronic diseases of the airways and other structures of the lung. ^¥^ Condition that affect blood flow and the blood vessels in the brain. ^Φ^ Serum creatinine >2.0 mg/dL prior to the hospitalization. ^Ψ^ Immunocompromised conditions including malignancy, having undergone organ transplantation, receipt of corticosteroids or immunosuppressive agent. ^Ω^ Any antimicrobial use (either oral or parenteral) within 3 months prior to the onset of index infection.

**Table 2 antibiotics-11-00354-t002:** Comparison of infection characteristics and antimicrobial therapy of patients in the pre- and post-implementation periods.

Variables	Total *n* = 1802 no (%)	Pre *n* = 901 no (%)	Post *n* = 901 no (%)	*p*-Value
**Sites of infection**				
Bloodstream infection	379 (21.03)	184 (20.42)	195 (21.64)	0.53
Catheter-related bloodstream infection	8 (0.44)	4 (0.44)	4 (0.44)	1.00
Urinary tract infection	430 (23.86)	204 (22.64)	226 (25.08)	0.22
Lower respiratory tract infection	725 (40.23)	401 (44.51)	324 (35.96)	<0.001
Gastrointestinal tract infection	267 (14.82)	122 (13.54)	145 (16.09)	0.13
Skin and soft tissue infection	194 (10.77)	108 (11.99)	86 (9.54)	0.10
Other	177 (9.82)	99 (10.99)	78 (8.66)	0.10
**Types of infection**				
Community-acquired infection	507 (28.14)	311 (34.52)	196 (21.75)	<0.001
Hospital-acquired infection	1295 (71.86)	590 (65.48)	705 (78.25)	
**Baseline vital signs** *				
Temperature, mean ± SD (°C)	38.52 ± 1.13	38.39 ± 1.14	38.65 ± 1.10	<0.001
Respiratory rate, mean ± SD (times/min)	23.33 ± 5.08	23.79 ± 5.26	22.87 ± 4.86	<0.001
Heart rate, mean ± SD (beats/min)	103.79 ± 20.91	103.87 ± 20.89	103.70 ± 20.95	0.89
MAP, mean ± SD (mmHg)	82.93 ± 17.52	82.86 ± 18.15	83.01 ± 16.87	0.86
**Laboratory results**				
Hematocrit, mean ± SD (mg%)	30.46 ± 7.01	30.80 ± 7.08	30.12 ± 6.93	0.04
White blood cell, median [range] * (1000/cu mm)	12.50 [0.005–135.14]	12.77 [0.005–108.48]	12.30 [0.02–135.14]	0.12
Serum creatinine, median [range] * (mg/dL)	1 [0–34]	1.19 [0.22–17.69]	1 [0–34]	0.39
**APACHE parameters**				
Having any organ insufficiency	558 (30.97)	247 (27.41)	311 (34.52)	0.001
Having acute kidney injury	540 (29.97)	292 (32.41)	248 (27.52)	0.02
ICU admission	178 (19.76)	157 (17.43)	335 (18.59)	0.20
On ventilator	910 (50.50)	470 (52.16)	440 (48.83)	0.16
Undergoing elective surgery	50 (2.77)	32 (3.55)	18 (2.00)	0.05
Undergoing emergency surgery	251 (13.93)	124 (13.76)	127 (14.10)	0.84
**Causative pathogens**				
*A. baumannii*	213 (11.82)	124 (13.76)	89 (9.88)	0.01
*E. coli*	325 (18.04)	153 (16.98)	172 (19.09)	0.24
*K. pneumoniae*	239 (13.26)	113 (12.54)	126 (13.98)	0.37
*P. aeruginosa*	165 (9.16)	89 (9.88)	76 (8.44)	0.29
*Enterobacter* spp.	41 (2.28)	17 (1.89)	24 (2.66)	0.27
*S. aureus*	37 (2.05)	23 (2.55)	14 (1.55)	0.14
*Enterococcus* spp.	48 (2.66)	25 (2.77)	23 (2.55)	0.77
Other Gram-negative bacteria	125 (6.94)	69 (7.66)	56 (6.22)	0.23
Other Gram-positive bacteria	56 (3.11)	30 (3.33)	26 (2.89)	0.59
**Initial prescription of targeted antimicrobials**				
Meropenem	894 (49.61)	436 (48.39)	458 (50.83)	0.25
Piperacillin/tazobactam	657 (36.46)	325 (36.07)	332 (36.85)	
Imipenem	32 (1.78)	17 (1.89)	15 (1.66)	
Others ^¥^	219 (12.15)	219 (12.15)	96 (10.65)	

* Vital signs at time of diagnosis of index infection. ^¥^ Other antimicrobials including Biapenem, Ceftolozane/tazobactam, Colistin, Sulbactam, Tigecycline and Vancomycin.

**Table 3 antibiotics-11-00354-t003:** Comparison of clinical outcomes of patients in the pre- and post-implementation periods.

Variables	Total *n* = 1802 no (%)	Pre *n* = 901 no (%)	Post *n* = 901 no (%)	*p*-Value
**Treatment outcomes**				
Favourable clinical response	1268 (70.37)	613 (68.04)	655 (72.70)	0.03
Favourable microbiological response *	256 (82.85)	159 (81.54)	97 (85.09)	0.43
	(*n* = 309)	(*n* = 195)	(*n* = 114)	
In-hospital mortality	361 (20.03)	173 (19.20)	188 (20.87)	0.38
ICU duration, mean ± SD, days	3.97 ± 9.48	4.79 ± 10.08	3.15 ± 8.76	<0.001
Duration of ventilator dependency mean ± SD, days	6.10 ± 11.33	7.21 ± 13.08	5.00 ± 9.13	<0.001
Fever duration, mean ± SD, days	6.78 ± 8.32	7.97 ± 9.97	5.59 ± 6.01	<0.001
Length of stay, mean ± SD, days	17.35 ± 17.23	18.90 ± 17.60	15.78 ± 16.72	<0.001
**Treatment complications**				
Bacterial superinfection	174 (9.66)	109 (12.10)	65 (7.21)	<0.001
Antibiotic-associated diarrhoea	36 (2.00)	20 (2.22)	16 (1.78)	0.50
**Days of antimicrobial therapy (DOT)**				
All antimicrobials	11.73 ± 10.99	12.22 ± 10.59	11.25 ± 11.36	0.06
Targeted antimicrobials	6.94 ± 5.55	7.64 ± 6.10	6.24 ± 4.84	<0.001
Ertapenem	0.57 ± 2.43	0.66 ± 2.83	0.47 ± 1.94	0.11
Meropenem	4.02 ± 5.55	4.32 ± 6.05	3.72 ± 4.98	0.02
Imipenem	0.14 ± 1.19	0.13 ± 1.27	0.15 ± 1.10	0.72
Piperacillin/tazobactam	2.22 ± 3.78	2.54 ± 4.17	1.90 ± 3.33	<0.001

* Only patients with a follow-up culture were included.

**Table 4 antibiotics-11-00354-t004:** Factors associated with favourable clinical response at the end of antibiotic therapy.

Variables	Unadjusted OR [95% CI; *p*-Value]	Adjusted OR [95% CI; *p*-Value]
Post-implementation	1.25 [1.02–1.53; *p* = 0.03]	127 [1.02–1.57; *p* = 0.03]
Underlying cardiovascular disease	0.68 [0.52–0.90; *p* = 0.006]	0.69 [0.52–0.92; *p* = 0.01]
Elective surgery	2.64 [1.18–5.91; *p* = 0.02]	2.57 [1.13–5.86; *p* = 0.03]
Immunocompromised status	0.71 [0.58–0.89; *p* = 0.002]	0.63 [0.51–0.80; *p* < 0.001]
Acute renal failure	0.41 [0.33–0.51; *p* < 0.001]	0.38 [0.31–0.48; *p* < 0.001]
*E. coli* infection	2.68 [1.95–3.69; *p* < 0.001]	2.60 [1.87–3.63; *p* < 0.001]
*A. baumannii* infection	0.66 [0.44–0.80; *p* = 0.001]	0.66 [0.48–0.90; *p* = 0.001]

OR—odd ratio, CI—confidence interval.

**Table 5 antibiotics-11-00354-t005:** Assessment of antimicrobial authorisation with the antimicrobial request and approval form during the post-implementation period.

Details of Antimicrobial Request and Approval Form	Number (%)
**Indication of targeted antimicrobial prescription**	
Specific therapy for MDR-bacterial infection	182 (20.20)
Empirical therapy	557 (61.82)
Having a contraindication of other antimicrobial agents	3 (0.33)
No documented indication	24 (2.66)
Other indications	14 (1.55)
**Targeted antibiotic approval**	
Pre-authorisation prescription	50 (5.55)
Post-authorisation prescription	851 (94.45)
Early discontinuation or discharge before evaluation	112 (13.16)
Post-authorisation prescription evaluation	739 (86.84)
Approval	665 (89.99)
No approval	74 (10.01)
**Additional recommendations**	
Dosage should be increased	2 (0.22)
Dosage should be decreased	5. (0.55)
Debridement is needed	34 (3.77)
Additional laboratory or imaging investigation is needed	30 (3.33)
Side effect should be monitored	19 (2.11)
Outpatient antimicrobial therapy should be offered	2 (0.22)
Intravenous to oral switching therapy should be offered	21 (2.33)

**Table 6 antibiotics-11-00354-t006:** Hospital infrastructure, existing antimicrobial stewardship activities, and targeted antimicrobial agents at three participating hospitals.

Variables	Sakaeo Crown Prince Hospital	Surin Hospital	Surat Thani Hospital
Geographical location	Eastern Thailand	Northeastern Thailand	Southern Thailand
Number of hospital beds	400	900	800
Number of qualified infectious disease physician	1	1	1
Antimicrobial stewardship team	Yes	Yes	Yes
**AMS interventions during the pre-implementation period**			
Antimicrobial request and approval form	No	No	Yes
Pre-prescription authorisation	No	No	Tigecycline
Post-prescription authorisation	No	No	Ertapenem Imipenem Meropenem
**AMS interventions during the post-implementation period**			
Antimicrobial request and approval form	Yes	Yes	Yes
Pre-prescription authorisation	Colistin	Ceftolozane/tazobactam Colistin Tigecycline	Biapenem Colistin Sulbactam Tigecycline
Post-prescription authorisation	Ertapenem Meropenem Piperacillin/tazobactam Vancomycin	Ertapenem Imipenem Meropenem Piperacillin/tazobactam Vancomycin	Ertapenem Imipenem Meropenem Piperacillin/tazobactam Vancomycin

## Data Availability

The study dataset is available from the corresponding author upon reasonable request.

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
