# Peer review of "Impact of Antibiotic Authorisation at Three Provincial Hospitals in Thailand: Results from a Quasi-Experimental Study"

_antibiotics, 2022, doi:10.3390/antibiotics11030354_

Round 1

Reviewer 1 Report

Antimicrobial stewardship in areas where few resources are available is important because these are often the areas with the greatest risk of over usage of antibiotics. This was a before and after intervention study. Comparison of before and after groups showed some significant differences but many factors were compared. Differences in urinary catheters and NG tubes should be explained. In Table 2 the p values do not align so the comparisons are not clear – is it applied to differences in overall wards with 2 degrees of freedom or type of wards.  There were several apparently favourable outcomes after implementation for example less bacterial superinfection and better response but how much may have been due to increased collateral attention by a microbiologist or pharmacist rather than just intervention. Covid-19 would also have affected prescribing strategy. The reasons for the changes observed need to be discussed as the authors are ascribing them to their intervention rather than confounders across the period of study. Were the AMS team able to achieve changes in treatment by the clinicians only by authorisation or by discussion when authorisation was not being requested.

Author Response

Please see the attachment (Page 2).

Reviewer 2 Report

General Comments

This is an important quasi-experimental study performed between Jan 2019 and December 2020 to determine the impact of customized antimicrobial authorization at three provincial hospitals in Thailand.  Nice study and very important to include literature on antimicrobial authorization studies in countries/lower resourced settings. 

Major critique: The methodology section needs to be between the introduction and the results.  It is very confusing to see the methods after the discussion.  Also, the Tables are out of order because the methods was placed after the discussion.

Abstract:

Important to include what the “favourable clinical response” that you are referring to here.  The abstract is the first intro to the manuscript and this was not clear. 

The decrease in the mean length of hospital stay may not be (and likely not) attributable to the days of therapy of antibiotics.  There are too many other confounders that must be taken into account besides antibiotics.

The last sentence in the abstract is excellent – succinct and really shows the importance of the manuscript. 

Introduction:

Overall the intro was succinct and easy to follow. 

7th line in introduction – change antimicrobial resistance to AMR (since you already defined in the first sentence of the introduction.  

2nd paragraph in introduction – consider changing to “most hospitals in Thailand had an ASP in place”

Results:

First sentence – consider “1,802 patients from each participating hospital…..”

Blood stream should be “bloodstream” (please correct this throughout the manuscript)

Page 5 and Table 4 – what do you mean by “bacterial superinfection”?  Please define this. 

Last sentence on Page 5 – is this 28-day in-hospital mortality?  If so, then please use the same variable within Table 4. 

Page 8 – I do not think that this sentence is accurate “However the potential confounder of COVID-19 should be minimal because most of the baseline characteristics of patients in both the pre- and post-implementation periods were similar”.  I would argue that despite similar patient characteristics, patients with COVID-19 possibly had fevers, abnormal chest X-rays (thus possibility of inappropriate diagnoses of bacterial pneumonias in patients who actually had COVID/viral pneumonia) and were likely prescribed more antibiotics than in pre-COVID groups.  Would consider removing or editing this sentence. 

Discussion

Page 7 - The sentence “On the basis of the study results, implementation of an antimicrobial authorization resulted in a higher rate of favourable clinical response, shorter ICU stays, shorter ventilation dependency duration, shorter fever duration, shorter hospital stays, and a lower rate of bacterial superinfection”. You actually cannot prove that it was the antibiotic recommendations that resulted in decreased hospital stays (there are too many confounders that could result in this).  Also, as stated above, I do not understand what you mean by “bacterial superinfection”. 

Methodology

Very clear and easy to follow.  Move Methods section to right after the Introduction and before Results section.  This was very confusing to see it after the Discussion

Table 1

Very clear and organized

Table 2

Can you please define what “cerebrovascular diseases” are? “respiratory tract diseases”, etc.  For example, pneumonia is a respiratory tract disease, but I assume that this is not what you are referring to.  Would be nice to have these definitions in a legend on Table 2. 

Also, what are “renal diseases”?  Do you mean “chronic kidney disease”?  If so, which stages? Please define these better in the legends. 

Define “immunocompromised host” in a legend. 

Change “HIV diseases” to “HIV” or “HIV disease”

Consider changing “Received medical devices” to “Underwent placement of catheter during hospitalization” 

Why are you including Nasogastric tubes?  This does not seem pertinent as they are rarely (if ever associated with infections).  Would consider removing this entirely from the manuscript unless you have a very good reason, or prior literature, pointing to its pertinence. 

Previous exposure to antimicrobial agent within 3 months.  Can you clarify whether these are oral antimicrobials?  IV antimicrobials?  Both?  Are you referring to only prior hospital exposure with these antibiotics?  This is not clear.  Consider a legend in the Table or defining better. 

Table 3

Baseline vital signs – Are these baseline vital signs on admission to the hospital or at time of diagnosis of infection?  Please clarify. 

Laboratory results – change “creatinine” to “serum creatinine”

Initial prescription of targeted antimicrobials – 12.15% are described as “others”.  This is a substantial proportion and it is important to include at the least the names of the “others” in a legend. 

Table 4

Define “targeted antimicrobials”

Author Response

Please see the attachment (Page 2).
